# Mammary Fibroadenoma in Cats: A Matter of Classification

**DOI:** 10.3390/vetsci9060253

**Published:** 2022-05-26

**Authors:** Filippo Torrigiani, Valentina Moccia, Barbara Brunetti, Francesca Millanta, Guillermo Valdivia, Laura Peña, Laura Cavicchioli, Valentina Zappulli

**Affiliations:** 1Department of Comparative Biomedicine and Food Science, University of Padova, 35020 Legnaro, Italy; valentina.moccia@phd.unipd.it (V.M.); laura.cavicchioli@unipd.it (L.C.); valentina.zappulli@unipd.it (V.Z.); 2Department of Veterinary Medical Sciences, University of Bologna, 40064 Bologna, Italy; b.brunetti@unibo.it; 3Department of Veterinary Sciences, University of Pisa, 56124 Pisa, Italy; francesca.millanta@unipi.it; 4Department of Animal Medicine and Surgery, School of Veterinary Medicine, Complutense University of Madrid, 28040 Madrid, Spain; edgargva@ucm.es (G.V.); laurape@vet.ucm.es (L.P.)

**Keywords:** feline, mammary, fibroadenoma, fibroadenomatous change

## Abstract

Benign mammary lesions are infrequent in cats. Among these, the most common is feline fibroadenomatous change, a hyperplastic/dysplastic change associated with hormonal imbalances. Although never thoroughly described in scientific literature, feline fibroadenomas, which share some morphological features with fibroadenomatous change, have been variably included in classification systems. The aim of this study was to characterise feline mammary fibroadenomas from a histological and immunophenotypical point of view in order to allow the standardisation of classification. Nine cases were retrospectively collected from eight female and one male cat with no history of hormonal stimulation. Diagnostic inclusion criteria were defined and immunohistochemistry was performed. Histologically, nodules were composed of neoplastic epithelial cells arranged in arborizing lobular-like structures surrounded by abundant proliferating stroma. In all analysed cases, epithelial elements showed immunolabelling for pancytokeratin, cytokeratin19, and β-catenin. Interestingly, five cases showed multifocal epithelial vimentin positivity. Epithelial nuclear oestrogen receptor positivity was observed in three of the nine samples. In all cases, myoepithelial cells did not extend into the interstitium. Stromal cells expressed vimentin, calponin, and mild β-catenin. The median Ki67 scores were 18% and 8.3% in the epithelial and stromal components, respectively. This study describes, for the first time, the morphological and immunophenotypical features of feline mammary fibroadenoma, highlighting its existence as a separate entity from fibroadenomatous change.

## 1. Introduction

Mammary tumours are common in cats, as they account for 17% of all neoplasms in this species and up to 25% of tumours in queens [1,2]. The majority of feline mammary tumours (FMTs) are malignant, with reported malignancy rates of 85–95% and with frequent aggressive behaviour [3]. FMTs are more commonly represented by simple carcinomas, tumours composed of a single neoplastic epithelial cell type [2,4]. Although infrequent, benign mammary lesions have also been reported in cats, and these include both benign neoplasms and hyperplastic/dysplastic lesions. Based on the most recent international classification from the David Thompson Foundation (herein DTF 2019) [4], the former includes simple, ductal, and intraductal papillary adenomas, whereas the latter includes duct ectasia, lobular hyperplasia, epitheliosis, papillomatosis, and fibroadenomatous change. Particularly, feline fibroadenomatous change (FAD change) is one of the most common non-malignant lesions of the feline mammary gland. It is the only fibroadenomatous lesion in the cat, composed of both stromal (“fibro”) and epithelial glandular (“adeno”) proliferation. It is also referred to as feline fibroepithelial hypertrophy, mammary fibroadenomatous hyperplasia, or feline mammary hypertrophy/hyperplasia/fibroadenomatous complex, and it accounts for 13 to 20% of mammary masses in this species [5]. Feline FAD change is a dramatic hyperplastic change which is induced by sex hormone stimulation. Young cycling or pregnant queens are more commonly affected, but also both female and male cats treated with sex-hormone therapy (i.e., megestrol acetate and medroxyprogesterone acetate) or with a history of endogenous sex-hormone increase (i.e., hormone-producing neoplasms) are affected [6,7,8,9,10,11]. Clinical signs include a rapid onset of mammary gland enlargement and frequent involvement of more than one gland in a diffuse or coalescing growth [5]. Interestingly, the lesion can regress spontaneously in young animals. The cornerstone of treatment of FAD change is the removal of hormonal influence (endogenous or exogenous), which usually induces the complete restoration of the mammary enlargement in 21–24 days [5]. Histologically, FAD change is composed of a proliferation of ducts and lobules designing a typical frond-like pattern within a variable amount of oedematous, collagen-rich, interlobular connective tissue. Ducts can be extremely ectatic and lobules are embedded and closely surrounded by a proliferating loose myxoid intralobular stroma [4,5,8,12]. The loss of epithelial cells’ polarity or mild cellular atypia can be observed and mitotic activity can be high in both the epithelial and stromal components [4,5].

FAD change is very common in the cat, but there is no clear evidence of its existence in the dog. Conversely, in the dog, there is a different “fibroadenomatous” lesion, which is a benign tumour classified as fibroadenoma by DTF 2019 [4]. Histologically, this entity is similar to the FAD change, but it usually presents as a single, well-circumscribed, uni-or multi-nodular mass, affecting one or rarely more mammary glands in the absence of hormone sensitivity [4]. In the previous classification of canine and feline mammary tumours published by the World Health Organization (herein WHO 1999) [13], feline fibroadenoma was included as a benign tumour, but was later excluded for insufficient reporting and difficulties in distinguishing the lesion from the more common feline FAD change [4,14].

The aim of this study was to better identify feline fibroadenoma versus FAD change in order to allow standardisation and comparison among studies.

## 2. Materials and Methods

Feline mammary gland samples classified as fibroadenoma according to the WHO 1999 classification were retrospectively collected from the archives of the Veterinary Anatomical Pathology Diagnostic Services of the Dept. of Comparative Biomedicine and Food Science (BCA Dept.) of the University of Padua (Padova, Italy), of the Dept. of Veterinary Science of the University of Pisa (Pisa, Italy), and of the Dept. of Animal Medicine and Surgery of the Complutense University of Madrid (Madrid, Spain). The collected data included the breed, age, sex, and neuter status. Histological evaluation of haematoxylin and eosin (HE) sections was performed by experienced veterinary pathologists with extensive knowledge in mammary gland histopathology (B.B., L.C., F.M., L.P., and V.Z.). Criteria for the diagnosis of fibroadenoma were a defined well-demarcated nodular lesion composed of a benign proliferation of both ductal/lobular epithelial elements and stromal component confined to a distinct mammary gland. The mitotic count of both epithelial and stromal components was separately calculated following recent guidelines as the sum of mitoses in 10 consecutive high-power fields (HPFs; diameter of the field of view = 0.55 mm; HPF = 0.237 mm^2^), starting the evaluation from the area with the most intense mitotic activity [15].

To characterise the cell populations and proliferation activity, immunohistochemistry (IHC) was performed on 4 μm serial sections of paraffin-embedded material mounted on either Superfrost Plus (Menzel GmbH, Braunschweig, Germany) or TOMO (Matsunami, Bellingham, WA, USA) microscope slides. A panel of mouse and rabbit monoclonal anti-human primary antibodies (Table 1), including anti-cytokeratin (CK) 19, anti-p63, anti-calponin (calp), anti-oestrogen receptor α (ER-α), and anti-Ki67, was applied to all samples. Additionally, due to limited tissue availability, anti-pancytokeratin (panCK), anti-β-catenin, and anti-vimentin (vim) antibodies were tested on 8/9 samples, whereas anti-cytokeratin 5/6 (CK5/6) was tested only on 3/9 samples. Finally, available anti-progesterone (PR) antibodies were tested on samples and controls to optimise new protocols. For panCK, CK5/6, β-catenin, vim, p63, calp, ER-α, PR, and Ki67, slides were processed using a semi-automated immunostainer (BenchMark, Ventana Medical System, Tucson, AZ, USA) at BCA Dept., University of Padua (Padova, Italy), following previously optimised protocols [16]. For CK19 and PR, a manual IHC protocol was also applied at the Department of Veterinary Medical Sciences, University of Bologna (Bologna, Italy) [17]. IHC for PR was also manually tested at the Dept. of Animal Medicine and Surgery of the Complutense University of Madrid (Madrid, Spain). Negative controls were obtained omitting the primary antibody; adjacent normal mammary gland, epidermis, dermis, and adnexa served as positive controls for Ki67, PanCK, CK5/6, CK19, β-catenin, calponin, vimentin, and p63; while normal feline ovarian and uterine tissue served as positive controls for ER-α and PR. IHC positivity was recorded as nuclear (Ki67, ER-α, PR, and p63), cytoplasmic (PanCK, CK 5/6, CK19, calp, and vim), and cytoplasmic/membranous (β-catenin) immunolabeling, distinguishing between epithelial and stromal components. The counting of 100 cells in 10 consecutive HPFs for a total of 1000 cells per sample was performed for nuclear antigens. No dual staining was performed.

## 3. Results

### 3.1. Study Population

Nine cases of feline mammary fibroadenoma were confirmed in nine different cats. Six of the cats were Domestic Shorthairs, one was Maine Coon, one was Abyssinian, and one was Persian, ranging in age from 1 to 10 years (mean 7 SD ± 2.8). In the study population, four of the nine affected cats were neutered females, four of the nine were non-pregnant intact females (one of which was 1 year old), and one of the nine was an intact male. No history of hormone treatment was indicated for all subjects, and no testicular or other tumours were reported for the male cat (Table 2).

### 3.2. Histology

All of the samples were characterised by single, moderately to densely cellular well-demarcated expansile nodules (Figure 1A) with a maximum size of 4.3 × 2.1 cm and a minimum size of 1.4 × 0.7 cm (Table 2).

Depending on the sample, the epithelial component represented from 20 to 80% of the nodules. Neoplastic epithelial cells were arranged in elongated ducts arborizing in lobular-like units of small ducts and tubules (Figure 1B). Multifocally, three of the nine samples showed large ectatic ducts often filled with amorphous, slightly eosinophilic material, sloughed epithelial cells, and foamy macrophages. The epithelial lining of ducts and tubules was composed of one to a maximum of three layers of irregularly polygonal cells, often surrounded by elongated basally located elements, mainly consistent with myoepithelial cells (Figure 1C) (see IHC results). Neoplastic epithelial cells were irregularly polygonal, up to 15 μm in maximum diameter, with mostly indistinct cell borders and a moderate amount of pale eosinophilic homogeneous cytoplasm (Figure 1C,D). The nuclei were round to oval, with coarse to finely stippled chromatin, and one or two prominent central nucleoli (Figure 1C,D). Anisocytosis and anisokaryosis were mild to moderate, with occasional macronuclei. The stromal component was divided into interlobular and intralobular stroma (Figure 1B) [4]. The interlobular stroma, between neoplastic lobular-like units, was consistently composed of thick and dense collagen bundles admixed with rare elongated cells with scant eosinophilic cytoplasm and spindle-to-cigar-shaped nuclei up to three to four microns in size. The interlobular stroma also showed disseminated small-calibre capillaries, mild multifocal oedema, and rare scattered inflammatory cells, mainly macrophages, lymphocytes, plasma cells, and mast cells. The intralobular stroma was closer to small ducts and tubules within and immediately around the lobular-like units (Figure 1B). It was composed of numerous spindle to stellate cells arranged in whorls and short bundles, and embedded in a moderately loose myxomatous matrix (Figure 1B). These cells often had scant cytoplasm and oval to elongated nuclei; occasionally, larger cells with more abundant cytoplasm and plump ovoidal to angular nuclei, up to 10 μm in diameter, were seen. In the stromal component, anisocytosis and anisokaryosis were overall mild, but more evident within the intralobular stroma.

Within the samples, the mitotic count in neoplastic epithelial cells (Figure 1D) ranged from 0 to 8 (mean 3.4 SD ± 2.78; median 3), while, in the stroma (interlobular and intralobular), the mitotic count ranged from 0 to 3 (mean 0.8 SD ± 1.05; median 1) (Table 2).

Mild multifocal, peripheral lymphocytic infiltration around the neoplasm was also observed in all samples. No additional lesions were observed in the surrounding mammary and cutaneous tissue.

### 3.3. Immunohistochemistry

The IHC results are summarised in Table 2. Cells (epithelial and myoepithelial) lining the duct and tubules consistently showed diffuse (100%) immunolabeling for panCK (Figure 2A), whereas CK19 was expressed only in the irregularly polygonal, mainly luminal epithelial cells (Figure 2B). The ductal and tubular elongated basally located cells were diffusely (100%) positive for p63 (Figure 2C) (except for one case, no. 2, which had discontinuous basal p63 positivity) and discontinuously (<100%) for calp (Figure 2D); the basally located cells also showed multifocal immunolabeling for CK5/6 and vimentin (Figure 3A). In six of the nine cases, both intra- and interlobular stromal cells showed multifocal cytoplasmic immunolabeling for calp (Figure 2D), while in two of the nine cases, calp-positive cells were detected only within the interlobular stroma. Only one case had no calponin-positive stromal cells (case no. 1).

Within the subset of analysed cases (eight), ductal and tubular neoplastic polygonal and basally located elongated cells showed diffuse membranous/cytoplasmic positivity for β-catenin (Figure 3B). In seven of the eight cases, unusual diffuse mild cytoplasmic positivity for β-catenin was observed in stromal spindle cells and in vascular endothelium, and it was more prominent in the intralobular stroma (Figure 3B). Nuclear localization of β-catenin was never observed. In the eight analysed cases, as expected, vimentin diffusely stained stromal spindle cells and the vascular endothelium. Interestingly, in five cases, cytoplasmic immunolabeling for vimentin was also observed in neoplastic polygonal epithelial cells, both in larger ducts (100% cells) and in lobular-like units (<100%) (Table 2), as well as in epithelial elements lining normal ducts in the unaffected adjacent mammary tissue.

The Ki67 score ranged from 2.3% to 34.2% (mean 16.4% SD ± 10.57%; median 18%) in epithelial cells, while in stromal cells, it ranged from 0.9% to 16.1% (mean 9.4% SD ± 5.89%; median 8.3%) (Figure 3C). Epithelial nuclear ER-α positivity was observed in three of the nine cases (Figure 3D)—the highest percentage (28.7%) was detected in the male subject. ER-α immunolabeling was not recorded in stromal cells.

The IHC for PR (three available clones) did not show robust positivity, neither in the positive controls (feline uterus and ovaries) nor in the samples, both under semi-automated and manual settings.

## 4. Discussion

The present study describes nine cases of feline mammary fibroadenoma, which, even if included in the WHO 1999 classification [13], has not been characterised in the literature before. In two previous studies, cell proliferation and telomerase expression were investigated in feline mammary gland lesions and, among others, FAD change and fibroadenoma were included; however, in both cases, the authors did not include the specific diagnostic criteria applied to differentiate the two entities [18,19]. According to WHO 1999 and the most recent classifications, the feline FAD change is a coalescing-to-diffuse not-well-demarcated hyperplastic change of the mammary glands associated with exogenous and endogenous sex hormone stimulation and sensitivity, whereas fibroadenoma is a well-demarcated nodular benign tumour not clearly associated with sex hormone stimulation [4,13,14]. However, the literature regarding the distribution in feline species and the features of these two “fibroadenomatous” lesions is confusing. The WHO 1999 includes the FAD change only in cats and fibroadenoma in both species, and describes that fibroadenoma is composed of stroma and epithelium, sometimes also admixed with proliferating myoepithelial cells [13]. In 2017, Goldschmidt and colleagues instead described FAD change in both species and fibroadenoma only in the dog [14]. Finally, the DTF 2019 mentions FAD change only in the cat, whilst fibroadenoma is described only in the dog [4]. In addition, the DTF 2019 authors further specify that lesions previously classified (WHO 1999) as feline fibroadenoma should be included as focal FAD change, since there is no evidence of a different biological behaviour and that canine fibroadenoma does not contain myoepithelial cells, thus being different from the more common canine complex adenoma.

In the present study, lesions diagnosed as feline fibroadenoma were collected and investigated from three different academic institutions. In our cases, feline mammary fibroadenomas overlapped the morphological features of canine fibroadenomas [4]. They were characterised by well-defined nodules composed of proliferating panCK+/CK19+ epithelial cells forming arborescent ducts and loose lobular-like units embedded in stroma in variable proportions. Myoepithelial (panCK+/p63+/vim+/calp+) cells were restricted to a single basal layer of ducts and tubules, confirming the lack of interstitial proliferation of myoepithelial elements. In all but one case, the stroma showed variable myofibroblastic differentiation detected by calponin expression. Calponin is a myoepithelial marker; however, it also stains myofibroblasts; therefore, more than one marker should be applied to differentiate these two cell types [20,21]. In our cases, the absence of positivity for cytokeratins and p63 allowed the identification of (vim+/calp+) myofibroblasts and not myoepithelial cells within the stroma, as described for canine fibroadenoma in DTF 2019. Interestingly, in five of the eight analysed cases, vim-positive epithelial cells were observed. This peculiar immunohistochemical feature has already been reported in the normal and neoplastic mammary glands of cats, but it has never been observed in other species [16,22,23]. While vimentin is a well-established marker of the negative prognostic process defined in cancer as epithelial-to-mesenchymal transition [24,25,26], little is known about its expression in non-neoplastic or benign tumoural epithelia. The co-expression of cytokeratins and vimentin in ductal epithelial cells of normal feline mammary glands has been interpreted and justified as presumably due to the presence of a putative, non-terminally differentiated luminal epithelial cell subtype diffusely distributed in ducts [16]. Although co-expression was not investigated in our study, we observed diffuse cytoplasmic positivity to panCK and CK19 in neoplastic and non-neoplastic epithelial cells, suggesting a co-expression with vim, as already reported [16,22,23]. In our cases, interestingly, in seven of the eight analysed samples, stromal cells had cytoplasmic positivity for β-catenin. β-catenin is a key component of intercellular junctions and is usually expressed on cell–cell boundaries in normal epithelia [27]. In various physiological and pathological situations, including cancer, β-catenin accumulates in the cytoplasm and eventually translocates to the nucleus, where it functions as a transcription factor [28,29]. In humans, nuclear and cytoplasmic β-catenin immunolabelling has been described in the proliferating stroma in different “fibroadenomatous” lesions of the breast [30,31,32,33]. More specifically, nuclear β-catenin expression was reported in breast fibroadenomas and mainly in benign phyllodes breast tumours [30,31,32,33]. Particularly in phyllodes tumours, the authors also reported cytoplasmic positivity [32]. These findings have led to the hypothesis that the epithelial component of fibroadenomatous tumours drives the proliferation of stromal elements via abnormal Wnt signalling [30]. Although our data are not sufficient to demonstrate a dysregulation of the Wnt signalling in feline mammary fibroadenomas, this similarity with the human counterpart suggests a potential alteration of this pathway also in cats. The Wnt signalling pathway has been implicated in the processes of vasculogenesis and angiogenesis in various physiological and pathological states [34,35]. In this regard, β-catenin expression in endothelial cells has been frequently observed during vascular remodelling and angiogenesis [36,37,38]. The β-catenin expression in the endothelial cells of stromal vessels in our samples is, therefore, not surprising and could be ascribed to stromal angiogenesis regulated by the Wnt signalling pathway.

In the past, fibroadenomas have been divided into low and high-cellularity subtypes [13,39], but the criteria and usefulness of this subdivision have been questioned and thus abandoned by the recent DTF 2019 [4]. In the feline cases presented here, no distinctive features were observed that would justify a subtyping of the lesion.

Furthermore, all cases were well-demarcated nodules affecting one mammary gland; therefore, they were considered more consistent with a benign tumour, rather than a hyperplastic change. No history of exogenous or endogenous sex hormone stimulation was reported in the subjects included in this study; they were mainly older than 4 years (seven of nine) and they were not pregnant when intact (four of nine), indicating an unlikely association with sex hormone stimulation in our cases. Moreover, the only male cat included in the present study had no previous history of hormonal treatment or of testicular or other tumours that could have suggested hormonal imbalance. Nevertheless, we investigated the expression of ER and PR, which are variably expressed in FAD change [12,40,41,42,43]. In our samples, ER-α immunolabeling was detected only in three of nine cases solely involving the epithelial component and with a percentage of positive cells higher than 20% (28.7%) only in the male subject. Unfortunately, PR expression could not be assessed in our sample set. The availability of anti-human PR antibodies cross-reacting in animal tissues is, at the moment, an unsolved issue. Previously cross-reacting antibodies are no longer commercially available and those tested did not give robust results after several attempted protocols [16,41,42,44,45,46].

In feline FAD change, ER-α positivity has been described as variable and usually restricted to epithelial elements. In one dated study, ER-α was not found in the cytosol or nuclei of two studied cases [7]. More recently, in FAD change, ER-α has been reported in 55.5% of cases (10/18) [12], 100% of cases (3/3) [39], and 12.5% of cases (1/8) [41,42], being significantly lower than normal glands or carcinomas [40]. In a single study evaluating ER-α in association with other markers in feline mammary neoplasia, the FAD change was included and described as variably positive without the indication of specific percentages [47]. Despite being based on a few studies, we can speculate that ER positivity is usually present in FAD change, but there is no threshold that can be used as a diagnostic marker. In a few studies, PR expression was found in up to 100% of FAD change cases [12,41,42,44].

Data regarding ER and PR expression in fibroadenoma in dogs and cats are, unfortunately, lacking. In dogs, only one report has described the development of a recurrent mammary fibroadenoma in association with the administration of progestogens which, however, underwent spontaneous regression after caesarean section [48]. No ER and PR expression was assessed. Based on the clinical setting, the recurrence, and the regression, the authors themselves stated within the discussion that, according to the most updated classification of canine mammary tumours [39] at the time of their study, the lesion should rather be classified as a FAD change than as a fibroadenoma.

To note, both mitotic activity and Ki67 expression in our cases were consistently higher in the epithelial component than in the stroma. High proliferative activity has been described in FAD change, as well as in fibroadenoma [13,18,47,49]. In one study, FAD change presented variable Ki67 positivity (20–60%) without distinction between epithelial and stromal cells [47]. In their study, Millanta and colleagues (2002) reported a higher Ki67 index in FAD change compared to normal mammary gland, mammary adenosis, and mammary in situ carcinoma, while the Ki67 index in FAD change was slightly, but not significantly, lower than that in invasive mammary carcinomas, the latter having a median Ki67 index of 18.7%. A differential count on stromal and epithelial elements in FAD change was not performed [49]. These results were confirmed in a following study [42]. Conversely, Dias Pereira and co-workers (2004) evaluated the Ki67 index in epithelial and stromal elements separately and found, in four samples of FAD change, a significantly higher percentage in the epithelial component (20.1%) compared to the stroma (15.1%). The mean Ki67 of the two low cellularity feline fibroadenomas included in their study had a significantly lower Ki67 index than FAD change in both epithelial (7.4%) and stromal (5.1%) components [18]. No additional data are available for canine fibroadenoma. In the cases of our study, the median Ki67 indexes were 18% and 8.3% in epithelial and in stromal cells, respectively. Data are generally too sparse to establish a mitotic or Ki67 cut off value that differentiates FAD change from fibroadenoma.

Based on these data, we can conclude that hormone receptors and proliferative activity are not significant markers to distinguish FAD change from fibroadenoma. The distinction can be based on morphological and clinical features and, since prognosis and treatments are different, the two entities should be maintained in both dogs and cats. Since the architectural and cytological features of the epithelial and stromal elements within the two lesions are overlapping, the distinction should be based on the extension and the demarcation of the lesion, and the involvement of either single or multiple mammary glands. In the case of a well-demarcated fibroadenomatous nodule affecting one gland, the diagnosis of fibroadenoma should be preferred. Conversely, when there is a poorly demarcated fibroadenomatous hyperplasia involving more glands, then classification as fibroadenomatous change would fit better. Clinical data, when available, might reinforce the distinction between fibroadenoma and FAD change. In this regard, FAD change has been consistently reported in young cycling queens or in cats with a history of endogenous or exogenous hormonal stimulation. A comment regarding the different nature and significance of the two lesions might accompany reports in which clinical data are not provided to the pathologist.

The authors believe that efforts in standardisation are not a waste of time, since otherwise, the comparison of scientific data and epidemiological studies will continue to suffer from classification biases.

## Figures and Tables

**Figure 1 vetsci-09-00253-f001:**
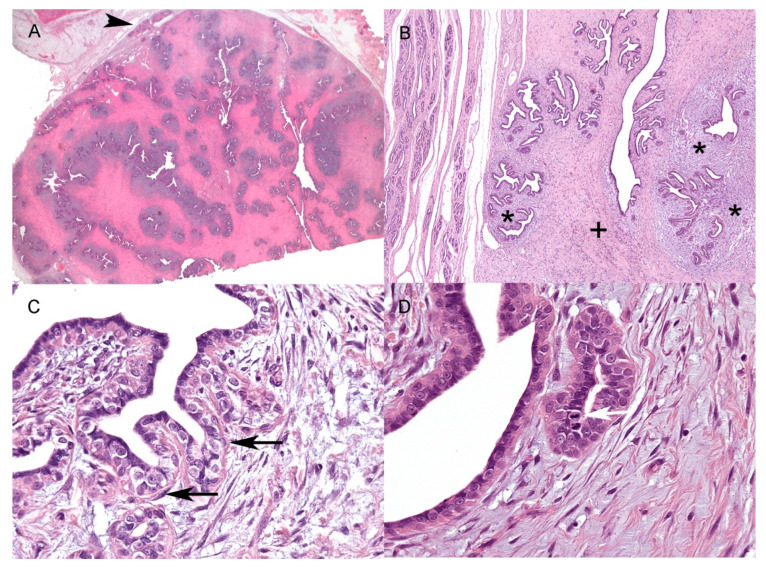
Histopathological features of feline mammary fibroadenoma. (**A**) Well-defined subcutaneous fibroadenoma with adjacent unaffected mammary gland (arrowhead) (size of the nodule at histology: 4.3 × 2.1 cm) (HE, 2×); (**B**) Unaffected mammary gland (on the left) and the lesion showing an elongated duct (centre of the lesion) and lobular-like structures. A more densely cellular, myxomatous intralobular stroma (asterisks) and a more collagenous, poorly cellular interlobular (+) stroma are evident (HE, 4×); (**C**) Elongated basally located cells (arrows), compatible with myoepithelial cells surrounding the irregularly polygonal epithelial cells (HE, 40×); (**D**) Frequent mitoses are present mainly within the neoplastic epithelial population (white arrow) (HE, 40×). HE = haematoxylin and eosin.

**Figure 2 vetsci-09-00253-f002:**
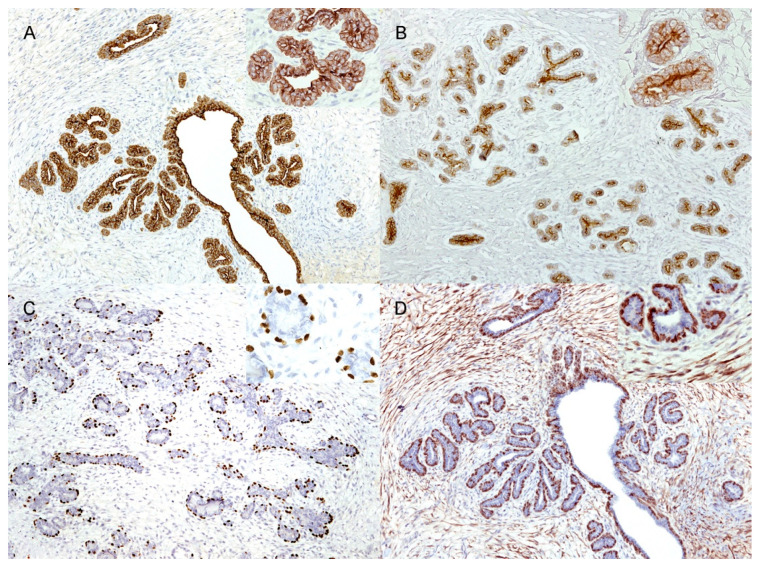
Immunohistochemical features of feline mammary fibroadenoma. (**A**) Neoplastic cells of ducts and tubules showing diffuse cytoplasmic positivity for pancytokeratin (10×). Inset: higher magnification (40×); (**B**) Polygonal ductal and tubular neoplastic epithelial cells exhibiting diffuse cytoplasmic immunolabelling for cytokeratin 19 (10×). Inset: higher magnification (40×); (**C**) Nuclear p63 positivity observed in basally located elongated cells consistent with myoepithelial cells (10×). Inset: higher magnification (40×); (**D**) Intra- and interlobular stromal cells, as well as ductal and tubular basally-located elongated cells showing cytoplasmic immunolabelling for calponin (10×). Inset: higher magnification (40×).

**Figure 3 vetsci-09-00253-f003:**
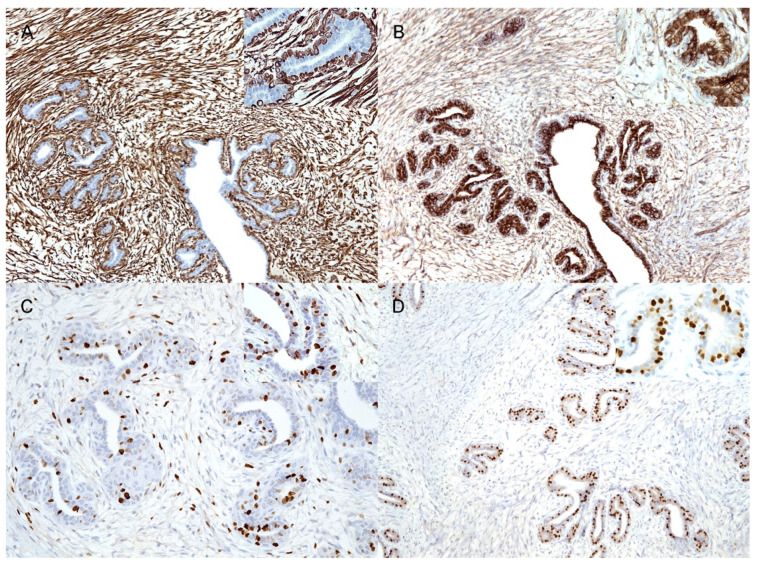
Immunohistochemical features of feline mammary fibroadenoma. (**A**) Stromal cells and basally located elongated ductal and tubular cells showing diffuse cytoplasmic positivity for vimentin (10×). Inset: higher magnification (40×); (**B**) Membrane/cytoplasmic positivity for β-catenin in neoplastic ducts and tubules as well as in the stroma (10×). Inset: higher magnification (40×); (**C**) Nuclear Ki67 immunoreactivity observed in both epithelial and stromal elements (20×). Inset: higher magnification (40×); (**D**) Nuclear ER-⍺ positivity restricted to neoplastic ductal and tubular cells (10×). Inset: higher magnification (40×).

**Table 1 vetsci-09-00253-t001:** Antibody panel for immunohistochemistry.

Antigen	Clone	Dilution	Host Species	Manufacturer
β-catenin	14/Beta-Catenin	1/100	Mouse	BD Biosciences
CK5/6	D5/I6 B4	1/50	Mouse	Invitrogen
CK19	BA 17	1/400	Mouse	Histo-line Laboratories
PanCK	AE1/AE3	1/100	Mouse	Dakocytomation
ER	ERa NCL-ER-6F11	1/40	Mouse	Novocastra
Ki67	MIB-1	1/50	Mouse	Dakocytomation
p63	4A4	1/200	Mouse	Santa Cruz Biotechnology
PR	PR88	1/80	Mouse	Biogenex
PR	1E2	prediluted	Rabbit	Roche
PR	PR16	1/80	Mouse	BioCare
Vimentin	V9	1/150	Mouse	Dakocytomation

CK, cytokeratin; PanCK, pancytokeratin; ER, oestrogen receptor; PR, progesterone receptor.

**Table 2 vetsci-09-00253-t002:** Cases included in the study: clinical, morphological, and immunohistochemical features.

Case no.	Breed	Age (yy)	Sex	Size *	MC ep	MC st	KI67 ep	Ki67 st	CK19 (C)	p63 (N)	Calponin (C)	ER 6F11 (N)	Vimentin ** (C)	β-catenin (M & C)	PanCK (C)
1	Abyssinian	10	F	2.5 × 2.2	3	0	14.5	8.3	100% ductal and tubular polygonal epithelial cells	100% ductal and tubular basal elongated cells	<100% ductal and tubular basal elongated cells	18.3% ductal and tubular polygonal epithelial cells	30% ductal and tubular polygonal epithelial cells	100% ductal and tubular cells	100% ductal and tubular cells
2	DSH	10	FS	2.7 × 2	0	0	3.1	2.3	100% ductal and tubular polygonal epithelial cells	<100% ductal and tubular basal elongated cell	<100% ductal and tubular basal elongated AND stromal cells	neg	no epithelial cells	100% ductal and tubular cells AND stromal cells	100% ductal and tubular cells
3	Maine Coon	3	FS	1.8 × 1.1	8	2	18.0	8.2	100% ductal and tubular polygonal epithelial cells	<100% ductal and tubular basal elongated cell	<100% ductal and tubular basal elongated AND stromal cells	neg	no epithelial cells	100% ductal and tubular cells AND stromal cells	100% ductal and tubular cells
4	DSH	5	FS	1.7 × 1.2	5	1	20.9	13.7	100% ductal and tubular polygonal epithelial cells	<100% ductal and tubular basal elongated cell	<100% ductal and tubular basal elongated AND stromal cells	neg	no epithelial cells	100% ductal and tubular cells AND stromal cells	100% ductal and tubular cells
5	DSH	5	FS	2 × 1.3	1	0	2.3	0.9	100% ductal and tubular polygonal epithelial cells	<100% ductal and tubular basal elongated cell	<100% ductal and tubular basal elongated AND stromal cells	neg	NA	NA	NA
6	DSH	1	F	2.8 × 1.9	3	1	34.2	14.6	100% ductal and tubular polygonal epithelial cells	<100% ductal and tubular basal elongated cell	<100% ductal and tubular basal elongated AND stromal cells	neg	ductal and tubular polygonal epithelial cells	100% ductal and tubular cells AND stromal cells	100% ductal and tubular cells
7	Persian	10	F	1.4 × 0.7	5	3	25.4	16.1	100% ductal and tubular polygonal epithelial cells	<100% ductal and tubular basal elongated cell	<100% ductal and tubular basal elongated AND stromal cells	17.6% ductal and tubular polygonal epithelial cells	90% ductal and tubular polygonal epithelial cells	100% ductal and tubular cells AND stromal cells	100% ductal and tubular cells
8	DSH	8	F	2 × 1	0	0	8,1	4.6	100% ductal and tubular polygonal epithelial cells	<100% ductal and tubular basal elongated cell	<100% ductal and tubular basal elongated AND stromal cells	neg	50% ductal and tubular polygonal epithelial cells	100% ductal and tubular cells AND stromal cells	100% ductal and tubular cells
9	DSH	8	M	4.3 × 2.1	6	1	21.5	15.7	100% ductal and tubular polygonal epithelial cells	<100% ductal and tubular basal elongated cell	<100% ductal and tubular basal elongated AND stromal cells	28.7% ductal and tubular polygonal epithelial cells	80% ductal and tubular polygonal epithelial cells	100% ductal and tubular cells AND stromal cells	100% ductal and tubular cells

DSH, domestic short hair; MC, mitotic count; ep, epithelial cells; st, stromal cells; C, cytoplasmic; N, nuclear, CK, cytokeratin, PanCK, pancytokertin; NA, not available; neg, negative. * Size is expressed as two major diameters per nodule in cm. ** Vimentin diffusely stained stromal fibroblasts, endothelial cells, and ductal and tubular basal elongated cells in all samples.

## Data Availability

Not applicable.

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
