# Peer review of "Mammary Fibroadenoma in Cats: A Matter of Classification"

_vetsci, 2022, doi:10.3390/vetsci9060253_

Round 1

Reviewer 1 Report

The study presents relevant data for pathologists to perform the classification of cases of feline fibroadenoma, especially for the differential diagnosis with feline fibroadenomatous change. As the objective of the study was to differentiate fibroadenoma from fibroadenomatous change, it would be very useful to present a table comparing the morphological and immunohistochemistry characteristics of these two pathologies.

The background and aims provided in the abstract are not fully clear.

L.16: “Similar to the latter for morphology”? Rewrite.

L.78: Just cases classified as fibroadenoma or also as feline fibroadenomatous change? How many cases were evaluated? Have any cases been reclassified?

L.357-359: Too vague sentence.

Author Response

Dear Reviwer,

we would like to thank you for your comment and your revisions.

We have considered your suggestion of presenting a comparison between fibroadenoma and FAD change in a table, and, although we agree that such tools are of great help in clarifying the main features and differences of two pathological entities, we are not sure wether this is applicable in our case. Indeed, from a strictly histological point of view, the only difference that we identified between fibroadenomas and FAD change is the presence of a single, well-demarcated expansile nodule in one mammary gland in the former, and the involvement of multiple glands in a coalescing, not demarcated fashion in the latter. The architectural and cytomorphological features of both epithelial and stromal components within the lesions are overlapping between fibroadenoma and fibroadenomatous change. Moreover, nor the immunohistochemical expression of steroid receptors, nor proliferative activity are sufficient to support one diagnosis or the other. In the light of these data, we believe that a table would not be as beneficial as in the case of two distinct pathological entities with strikingly different histological and immunohistochemical features. Nevertheless, based on your comment we have changed the discussion at lines 352 - 355.

In the following we provide a point-by-point response to your comment.

The background and aims provided in the abstract are not fully clear.

Thank you for your comment, we have changed the background and aim sections in the abstract according to your suggestion.

L.16: “Similar to the latter for morphology”? Rewrite.

Thank you for your comment, we have changed the sentence in order to better clarify the concept.

L.78: Just cases classified as fibroadenoma or also as feline fibroadenomatous change? How many cases were evaluated? Have any cases been reclassified?

Thank you for your comment. We hope we understood your suggestion correctly as referred to line 78 describing the differences between 1999 WHO and the new CL Davis classifications. The 1999 WHO classification of feline mammary tumours included fibroadenoma as a separate entity from the fibroadenomatous change, although in scientific literature fibroadenoma had not been described in this species. In the light of this lack of data, the recent CL Davis classification has excluded feline fibroadenoma, while feline fibroadenomatous change has been kept within the hyperplastic/dysplastic disorders. In these regards no cases were reevaluated and reclassified by the authors of CL Davis fascicle.

L.357-359: Too vague sentence.

Thank you very much for your comment, we have better specified the conditions in which FAD change has been consistently described.

Reviewer 2 Report

General comment
The manuscript is very well written and the structure of Introduction, Materials and Methods, Result, and Discussion is systematic and easy to read. I would also like to see the possibility of genetic testing as an issue for future research.

Line121-126
Can you show the distribution of the location of the mammary glands where the lesions occurred? It would be interesting to know the trend.
Fig1,2
It would be easier to understand if you could display a bar that shows the size of the lesions.

Author Response

Dear Reviwer,

we would like to thank you for your comments and your revisions. Your suggestion of performing genetic tests for future studies is surely intriguing and would help clarifying the pathogenesis of lesions such as fibroadenoma in which two distinct components undergo neoplastic proliferation.

In the following we provide a point-by-point response to your comments.

Line121-126
Can you show the distribution of the location of the mammary glands where the lesions occurred? It would be interesting to know the trend.

Thank you very much for your comment, we agree that this information could provide additional and useful information on this pathologic entity, this information however was available only for two cases. We tried to contact the referring veterinarians but none were able to provide this information nor they had specified it in the request form when they submitted the samples for histopathological evaluation. So unfortunately we are unable to provide the complete information to describe the trend.

Fig1,2
It would be easier to understand if you could display a bar that shows the size of the lesions.

Thank you for your suggestion. Images in Figure 1 and 2 were taken with a high definition slide scanner (D-Sight, A. Menarini diagnostics S.r.l., Italy). This instrument provides high quality digital images but unfortunately does not allow to insert scale bars in the picture. We also believe that placing a scale bar with an image editing program after the acquisition would have been inaccurate and not representative of the original size. Since also the journal does not require the scale bar we hope that magnification stated as the objective used for capturing the image gives enough information on what is shown, and for figure 1A we included in the figure legend the size in cm as measured on the histological slide (lines 139-140).